# Egg Rejection and Nest Sanitation in an Island Population of Barn Swallows (*Hirundo rustica*): Probability, Response Latency, and Sex Effects

**DOI:** 10.3390/ani12213027

**Published:** 2022-11-03

**Authors:** Qiuhui Yang, Xiangyang Chen, Ziqi Zhang, Jingru Han, Neng Wu, Canchao Yang

**Affiliations:** Ministry of Education Key Laboratory for Ecology of Tropical Islands, College of Life Sciences, Hainan Normal University, Haikou 571158, China

**Keywords:** avian brood parasitism, nest cleaning, parasitic cuckoo, predation risk, reproduction success

## Abstract

**Simple Summary:**

Most birds build nests for laying eggs and rearing their offspring. They frequently clean their nests by removing foreign objects, such as leaves, small branches, stones, and feces. Further, some birds recognize and remove foreign eggs deposited by brood parasites. Nest sanitation and egg rejection are related behaviors as both impact brood survival and involve an accept/reject decision regarding the content of the nest. Here, we examined these behaviors in barn swallows (*Hirundo rustica*), one of the most abundant and widespread birds in the world. The results suggest that nest sanitation may be a more ancient behavior because it occurs more frequently, and foreign objects were removed sooner upon discovery than egg rejection. Therefore, nest sanitation may constitute an evolutionary precursor to foreign egg rejection; however, nest sanitation rarely increases egg rejection regarding probability and response latency. Female and male swallows engaged in nest sanitation and egg rejection, implying that both sexes are affected by this type of natural selection.

**Abstract:**

Bird nests function as vessels for eggs and nestlings, and an environment for rearing offspring. However, foreign objects falling into bird nests and nestling eggshells may be harmful. Moreover, the smell of fecal sacs increases the risk of detection by predators. Many bird species have evolved nest sanitation to prevent damage to their nests. Furthermore, egg rejection evolved in some birds to thwart brood parasites that lay eggs in their nests. We studied 133 nests of barn swallows (*Hirundo rustica*) in an island population through a nest content manipulation experiment to determine nest sanitation and egg rejection behaviors and their relationship. Swallows rejected non-egg foreign objects more frequently (100% vs. 58.6%) and sooner than parasite eggs, which supports the hypothesis that nest sanitation is a pre-adaptation to egg rejection. However, nest sanitation did not increase egg rejection, either in probability or latency. Furthermore, both sexes incubated the eggs, cleaned the nests, and removed parasite eggs, implying that both are confronted with natural selection related to nest sanitation and brood parasitism. However, females invested more time in these behaviors than males. This provides evidence for the evolutionary relationship of nest sanitation and egg rejection behaviors in barn swallows.

## 1. Introduction

Most birds build nests to lay eggs and to rear their offspring [1], and nests are key elements that determine reproductive success. Therefore, many bird species have evolved various behaviors to optimize their nests. For example, nest sanitation has evolved as an adaptation to improve reproductive success [2,3]. Nest sanitation by bird parents includes the removal of foreign materials, such as objects that have fallen from above the nests (such as leaves, branches, and stones), egg shells after hatching, and the feces of the nestlings [4]. Such cleaning behavior reduces the risk of olfactory detection by predators and may avoid damage to the eggs and harm to the nestlings caused by hard or sharp objects [1,4].

Comparable to nest sanitation, egg rejection by bird parents also constitutes the removal of foreign objects from nests; however, this behavior is a highly specific adaptation that has evolved as a defense against brood parasitism [5]. Avian brood parasites lay eggs in the nests of other birds (conspecific or heterospecific hosts) and thus transfer the parental costs of rearing their offspring to other individuals, which led to the evolution of egg rejection behaviors [6,7]. In the case of obligate brood parasites such as the common cuckoo (*Cuculus canorus*), the cost to the hosts is very high because of the loss of a breeding cycle or even an entire breeding season [8]. Hence, brood parasitism acts as a selection pressure resulting in various anti-parasitism defenses, among which egg rejection is one of the most important and effective strategies [9,10].

Nest sanitation and egg rejection share behavioral similarities; however, the cognitive levels involved in these two behaviors should differ. For nest sanitation, bird parents need only distinguish non-egg objects from eggs. However, for egg rejection, they must discriminate parasite eggs from their own eggs through subtle traits, such as color [11], markings [12], shape [13], and size [14]. Thus, nest sanitation is considered a pre-adaptation to egg rejection [15,16], not only due to the lower cognition level required but also because it is more common. All bird species that build nests are presumed to exhibit nest sanitation, while only some species (i.e., the hosts) face parasitism pressure.

In a previous study, the pre-adaptation hypothesis was supported by the results obtained after testing the relationship between nest sanitation and egg rejection in barn swallows (*Hirundo rustica*) [16]. This study found a strong positive correlation between nest sanitation and egg rejection probability across different geographic populations, and nest sanitation probability was consistently higher than egg rejection probability, in each population [16]. This indicates that nest sanitation is an original (i.e., ancient) behavior that may act as a pre-adaptation to egg rejection. Nevertheless, several questions remained to be addressed, for instance, if nest sanitation is simpler than egg rejection, upon discovery, would birds recognize non-egg objects sooner than foreign eggs? Are there sex-specific differences in nest sanitation and egg rejection?

To answer these questions, we studied nest sanitation and egg rejection behaviors in barn swallows using nest content manipulation experiments with a large sample size (133 nests) and video monitoring (>187,200 min). Based on the pre-adaptation hypothesis, we predicted that nest sanitation would be executed sooner upon discovery than egg rejection in the same nest. We also predicted that males and females would participate in nest sanitation and egg rejection if both sexes were engaged in egg incubation. Moreover, because previous studies proposed that nest sanitation would elicit egg rejection more frequently, the promotion effect of nest sanitation on egg rejection was also investigated.

## 2. Materials and Methods

### 2.1. Study Site and Study Animals

The study site was located in the central area of Hainan Island (19°20′ N–20°10′ N, 108°21′ E–111°03′ E), which is the second largest island of China, with an area of 33,900 km^2^. Hainan Island is located to the south of China and is characterized by sub-tropical and tropical climates, with an average temperature of 22.5–25.6 °C and 1500–2500 mm annual precipitation [16,17]. The central area consists of large, continuous natural forests, with some small villages scattered among them. This study was conducted in these scattered villages, in which barn swallows build nests under the eaves of residential houses. The barn swallow is the most abundant and widespread species of the Hirundinidae family, which is distributed across North America, Europe, Asia, and North Africa [18,19]. Their total population is estimated to be 1.1 billion, thus representing the fourth most abundant bird species [18]. In China, the parasitism rate of barn swallows varies from 0% to 2.4%, with the common cuckoo being the parasite [20].

### 2.2. Data Collection

We searched for swallow nests along the eaves of residential houses in the study area from late February 2021, when the birds began to arrive and reproduce. Active nests were examined regularly (approximately once every three days) to monitor reproductive processes. When laid eggs were found, the nests were monitored daily to record the day of incubation initiation. The nest content was manipulated by placing a blue mock egg (hereafter termed mock trial, *n* = 63) or a blue mock egg plus a half peanut shell (hereafter mock + peanut trial, *n* = 70) in each observed nest (Figure 1). Each nest received only one of these treatments. The mock egg represented a parasite egg, whereas the peanut shell half represented a non-egg foreign object. The mock egg was made of polymer clay; it had the size of a swallow’s egg but had a slightly greater mass [16]. However, the swallow is a grasp rejector and a minimal mass difference did not affect its rejection behavior [21]. After foreign objects were inserted, a mini-video camera (WJO3, Hisilicon, Shenzhen, China) was mounted to monitor the nests for 24 h. Thereafter, the camera was removed, and the nests were monitored by an observer every 24 h until the sixth day (from nest content manipulation). The camera was used for 24 h monitoring for each nest rather than for six days because most responses occurred within 24 h [20] and the workload of analyzing such large amounts of video material was not manageable. To obtain a large sample size and ensure experiment time and precision, several observers were scattered among the study site to investigate swallow nests and perform experiments during the same period. To avoid pseudo-replication of sampling nests from the same parental birds, the experiment was conducted on nests with an egg-laying date (the date of the first laid egg) within 38 days, i.e., the number of days between the first and last nests that were used for the experiment was 38. Because the main breeding cycle (from the beginning of egg incubation to the fledging) of barn swallows lasts approximately 38 days, limiting the duration of the experiment would prevent pseudo-replication [20]. This is one of the reasons why seven days of video monitoring for each nest was too large a workload, because a large sample size was acquired within a limited period of time. The videos were played back in the laboratory to identify rejection cases and to record the sex of the birds that performed the rejection. Although the barn swallow does not have obvious sex dimorphism, males have slightly brighter plumage and longer wings and tails than the females [22]. Such differences could be compared and detected in the video record of nests because both males and females incubated the eggs and often appear on the screen at the same time. If no rejection response occurred within 24 h, the investigation by observers every 24 h until the sixth day was used to determine which and when a behavioral response occurred. If a mock egg and/or peanut shell was punctured or disappeared on the day of checking, while the eggs were still incubated by the parental birds, it was considered a rejection. In contrast, if the foreign objects were found in the nests and incubated by the parental birds after six days of investigation, it was considered acceptance [16].

### 2.3. Statistical Analyses

The response latency data for nest sanitation and egg rejection were accurate to 1 min within 24 h and to 1 d after 24 h until the sixth day. Numeric data were transformed to ranked variables for further analyses. The response latency between nest sanitation and egg rejection in the same nest (i.e., mock + peanut trial) was paired and compared using Wilcoxon signed-rank tests; Wilcoxon rank-sum tests were used to compare the response latency without paired variables. To elucidate the promotion effect of nest sanitation on egg rejection, the probability of egg rejection between the mock trial and mock + peanut trial was compared using a generalized linear mixed model with the Markov Chain Monte Carlo (MCMCglmm) method. In the MCMCglmm model, the egg rejection probability was the response variable; the treatment (i.e., mock trial or mock + peanut trial), egg laying date (date of the first laid egg for each nest), and clutch size were the fixed effects; and nest identity was included as a random effect. The interaction between treatment and egg laying date or clutch size was also tested. Fisher’s exact test was used to compare the ratio of females and males exhibiting egg rejection or nest sanitation. The Wilcoxon signed-rank test and Wilcoxon rank sum test were performed using the *MASS* and *coin* packages, and the MCMCglmm model was fitted using the *MCMCglmm* package in R (version 4.1.0) for Windows (https://www.r-project.org/; accessed on 11 September 2022). All statistical tests were two-tailed, and statistical significance is reported at *p* < 0.05.

## 3. Results

The results of response probability indicated that the egg rejection rates of the mock and mock + peanut trials were 49.2% (*n* = 63) and 58.6% (*n* = 70), respectively (Table 1). Birds treated with mock egg + peanut shell removed the peanut shell in 100% of the cases (*n* = 70). The results of response latency showed that nest sanitation latency was significantly lower than egg rejection latency (Z = 547.5, *p* < 0.001, Wilcoxon signed-rank test; Figure 2a). However, for the nest sanitation latency between acceptance and rejection of the egg model in the mock + peanut trial (Figure 2b) and the egg rejection latency between the mock and mock + peanut trials (Figure 2c), no statistical significance was detected (W = 731.5 and 757, *p* = 0.103 and 0.169, respectively; Wilcoxon rank sum test). According to the video record, all rejections of either the mock egg or the peanut shell were executed by grasping rather than puncturing, and no pecking marks were found on mock eggs or peanut shells in the acceptance cases. No desertion of nests was observed. Males and females exhibited egg incubation, egg rejection, and nest sanitation. However, more than 50% of rejection behaviors were performed by females, in all trials (Figure 3). In the mock + peanut trial, females and males exhibited 64.3% and 17.9%, respectively (*p* = 0.069, Fisher’s exact test), and both sexes engaged in rejection in the remainder (Figure 3). Regarding sex-specific differences, either the female performed egg rejection while the male performed nest sanitation or vice versa. In the rejection cases of the mock trial, both sexes participated in egg rejection, with females and males exhibiting 57.9% and 36.8%, respectively (*p* = 0.738, Fisher’s exact test; Figure 3). In one case (5.3%), both partners engaged in egg rejection as the male grasped the mock egg to remove it from the nest cup and the female removed it further. With respect to nest sanitation behavior, females performed 90.5% of the cases, whereas males acted in only 9.5% of cases (*p* < 0.01, Fisher’s exact test; Figure 3). According to a recent suggestion of *p*-value classification by Muff et al. [23], the MCMCglmm indicated that there was weak evidence that the treatment predicted the egg rejection probability (*p* = 0.054). Furthermore, neither the egg laying date, clutch size, nor the interaction of treatment × laying date and treatment × clutch size predicted the rate of mock egg rejection (Table 2).

## 4. Discussion

According to our results, swallows recognized and rejected mock eggs in 49.2% of the cases. This egg rejection rate was approximately 10% higher compared with the cases in which a peanut shell was placed in the nest at the same time. However, the difference was close to but did not achieve statistical significance (*p* = 0.054). This result thus provides weak evidence to support the hypothesis that nest sanitation increases the probability of egg rejection behaviors [21], which was confirmed in some previous studies [24] but rejected in others [25,26,27,28]. Furthermore, our results indicated that egg rejection latency seemed to be slightly lower compared with the situation in which nest sanitation occurred; however, no significant difference was found. Therefore, the current study showed that nest sanitation rarely elicits an egg rejection response more frequently or sooner. We propose that population-specific differences explain these differences best. The current study was performed on a different swallow population, which was located in the southernmost part of China, with a straight-line distance of >3300 km from the northern population used in the previous study [16]. Further, previous studies have indicated that egg rejection rates differ between populations because of differences in instant or potential parasitism risk or historical contact with parasites [10,29,30]. The population of Hainan Island in this study was exposed to higher species diversity of parasitic cuckoo than the northern population [31], and no cases of natural parasitism were observed in either of these two populations. However, this does not indicate that the intensity of selection pressure from parasitism is similar between these populations because historical contact between parasites and hosts is responsible for the long-term maintenance of egg recognition in hosts [29,32,33]. Therefore, the level of egg recognition may be regarded as a more suitable way to evaluate host response to selection pressure. Consistent with the fact that parasitic cuckoos on Hainan Island are much more diverse than in the area of the northern population [31], the egg rejection rate of the Hainan population was twice that of the northern populations (49.2% as opposed to 24% in the northern population [16]). This implies that the Hainan population is more sensitive to parasitic eggs. As a result, if nest sanitation promotes egg rejection to some extent, the northern population would show increased sensitivity to parasite eggs (the egg rejection rate was twice as high; 56% [21]), while the Hainan population had much less space for sensitivity increase (49.2% to 58.6% in the present study). Interestingly, the rejection rates after promotion were similar between populations (56% vs. 58.3%), which implies that different barn swallow populations have a similar limitation in sensitivity to parasite eggs, and this limitation may reflect their interaction status with parasites in history. Barn swallows are generally migratory [34]; however, it is unknown whether gene flow occurs between the northern and Hainan populations. According to our personal observations, some barn swallows were residents of Hainan Island. The difference in population distance and egg rejection rate implies that they are unlikely to be from the same source as winter migrators. Thus, the northern and Hainan populations were presumed to be distinct. Further studies considering migration tracking and gene flow analyses are required to confirm the relationship between these populations. Additionally, the larger sample size of the current study may in part explain the discrepancies regarding egg rejection promotion rates: the previous study used 62 nests [21] while in the current study, 133 nests were examined. However, considering that the sample size in the previous study is commonly considered sufficient for studies on egg rejection, e.g., [35,36,37,38,39], and the rejection rates were stable between different previous studies for the northern population [16,21], the larger sample size may not sufficiently explain the discrepancies.

The result of response latency between egg rejection and nest sanitation in the same nest illustrated that the parental birds executed nest sanitation much sooner than egg rejection upon discovery. This is consistent with one of our predictions and, to some extent, supports the pre-adaptation hypothesis. Nest sanitation and egg rejection behaviors are performed by removing foreign objects from nests, but they require different cognitive levels. Furthermore, nest sanitation is more common and influences both the egg incubation and nestling feeding stages during reproduction, whereas egg rejection is a more specific behavior focusing on host species and individuals and influences egg incubation or even the early period of egg incubation [40,41,42,43]. If a cognitive behavior is more original, it is reasonable to predict that cognition would be easier to achieve when compared with a later evolved behavior. Our study showed that swallows recognized and rejected non-egg objects more frequently and sooner than mock eggs, thus providing evidence for this assumption. Nevertheless, to support the pre-adaptation hypothesis with unambiguous and stronger evidence, the sensory connection between nest sanitation and discrimination of foreign eggs must be examined.

According to the video recordings, female and male partners incubated their eggs and performed nest sanitation and egg rejection. This indicates that respective selection pressure affects both sexes; however, females rejected mock eggs and non-egg objects more frequently than males, and females invested more time in egg incubation than males, while males would also take responsibility for surveillance and guarding.

## 5. Conclusions

In summary, we conclude that: (1) nest sanitation does not promote the occurrence of egg rejection in our studied population of barn swallows, and the difference in the promotion effect of nest sanitation on egg rejection may depend on the difference in sensitivity to parasite eggs between populations; (2) nest sanitation is an original behavior and may act as a pre-adaptation to the egg rejection behavior in barn swallow; and (3) both the male and female barn swallow have evolved cognitive capacity to recognize non-egg objects and parasite eggs because both sexes are under the selection pressure of reproduction loss related to nest sanitation and brood parasitism.

## Figures and Tables

**Figure 1 animals-12-03027-f001:**
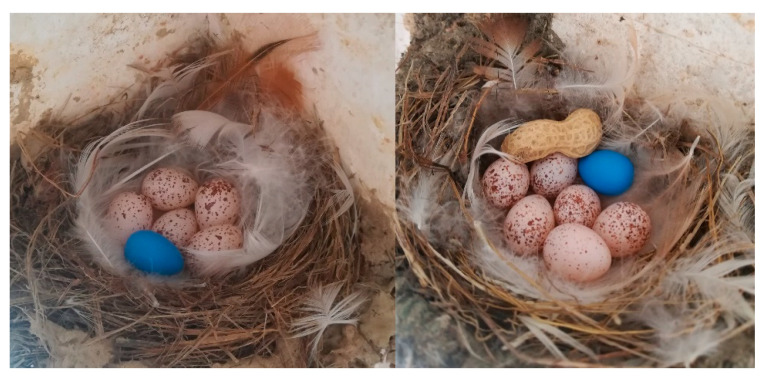
An example of mock trial (**left**) and mock + peanut trial (**right**) in this study.

**Figure 2 animals-12-03027-f002:**
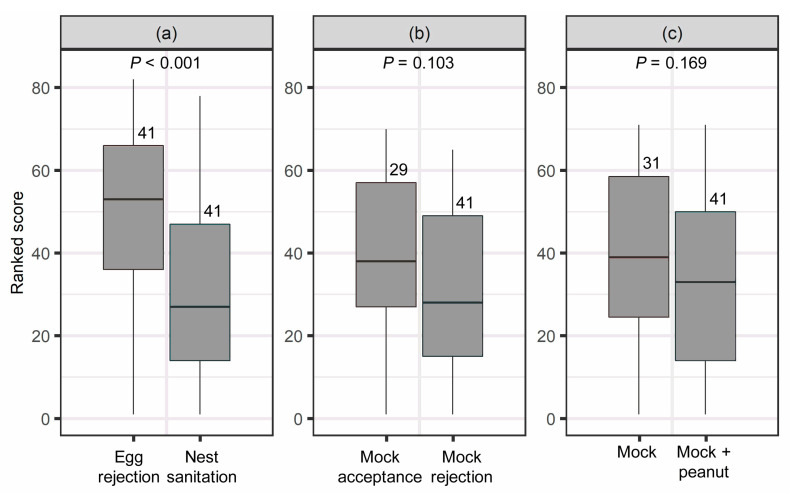
Boxplots (median, quartiles, and range) for the response latency of egg rejection and nest sanitation in barn swallows. (**a**) Comparison of response latency between paired egg rejection and nest sanitation in the mock egg + peanut shell (model + peanut) trial. (**b**) Comparison of nest sanitation latency between mock acceptance and mock rejection subsets in the mock + peanut trial. (**c**) Comparison of egg rejection latency between mock and mock + peanut trials. Numbers above boxes indicate the sample size (i.e., number of nests).

**Figure 3 animals-12-03027-f003:**
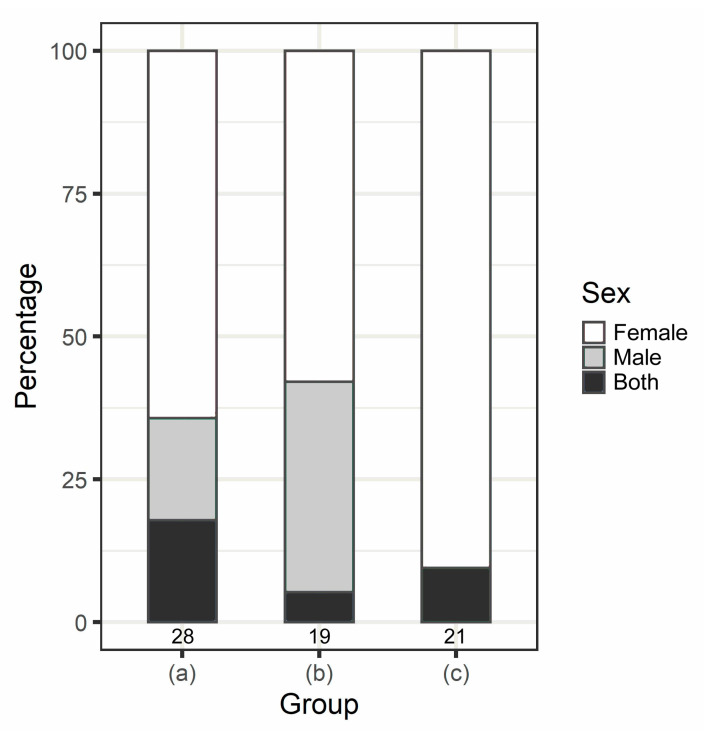
Percentage of egg rejection and/or nest sanitation performed by male and/or female barn swallows. (a) Mock egg + peanut shell trial: egg rejection and nest sanitation occur in a single nest. In this group either female or male performs both egg rejection and nest sanitation, or both sexes participate by each sex performing one of the behaviors (egg rejection or nest sanitation) without overlap. (b) Mock egg trial: only egg rejection occurs. In this group, both partners may cooperate to reject the mock egg. (c) Mock egg + peanut shell trial: only nest sanitation occurs. Numbers under bars indicate the sample size (i.e., number of nests).

**Table 1 animals-12-03027-t001:** Probability of mock egg rejection and/or nest sanitation in the mock egg trial (mock) and mock egg plus peanut shell trial (mock + peanut).

Response to Foreign Object		Treatment	
		Mock	Mock + Peanut
Mock egg	Rejection	31 (49.2%)	41 (58.6%)
	Acceptance	32 (50.8%)	29 (41.4%)
Peanut shell	Rejection	-	70 (100%)
	Acceptance	-	0 (0%)
Number of nests		63	70

**Table 2 animals-12-03027-t002:** Egg rejection probability of barn swallows following experimental treatments and egg laying date from generalized linear mixed models by Markov Chain Monte Carlo technique. Cr. I.–critical interval.

Fixed Effect	Posterior Mean	Lower 95% Cr. I.	Upper 95% Cr. I.	*p*
Intercept	0.760	−0.265	1.699	0.130
Treatment *	0.621	−0.017	1.205	0.054
Egg laying date	0.006	−0.015	0.026	0.574
Clutch size	0.027	−0.151	0.247	0.810
Treatment × Egg laying date	−0.003	−0.016	0.010	0.616
Treatment × Clutch size	−0.013	−0.150	0.100	0.840

* Treatment refers to the mock egg and mock egg + peanut shell trials. Nest identity was included as a random effect.

## Data Availability

The data presented in this study are available on request from the corresponding author.

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
