# Peer review of "Egg Rejection and Nest Sanitation in an Island Population of Barn Swallows (*Hirundo rustica*): Probability, Response Latency, and Sex Effects"

_animals, 2022, doi:10.3390/ani12213027_

Round 1

Reviewer 1 Report

Review for

Egg rejection and nest sanitation in an Island population of barn

swallows (Hirundo rustica): frequency, response latency, and sex effects

animals-1985135

In this study Yang et al. perform a clever experiment on barn swallow (Hirundo rustica) nests, which is an occasional host of the brood parasitic common cuckoo (Cuculus canorus). The experiment consists of placing dummy eggs with or without a peanut shell in 133 swallow nests, to test nest sanitation and egg rejection rates in the studied species. The manuscript is well written, clear, and easy to read, and it contributes to the body of knowledge regarding egg rejection decisions. I have only some easily addressable issues regarding the manuscript, which I listed bellow.

Major issues:

L100-133: There is no information in the Data collection section about the sample size of the experiment, the sample sizes of the experimental groups, how they were distributed to one of the two groups. I found the sample sizes in Table 1, but that is only on page 4, so it should be mentioned in the main text also.

L100-133: Egg mass might alter rejection decisions. Authors should state the mass of the mock eggs and if it differed significantly from the mass of barn swallow eggs. If there is a significant difference in mass, authors should address this in the Discussion.

L100-133: How the sex of the birds was identified and how consistent is the sex identification is missing from the text. In a species with a high degree of similarity between sexes, the repeatability of sex identification should be tested and mentioned.

L100-133: I would be happy to see a picture of the eggs and peanut shell, either in the nest or outside of it. I think adding this would aid the understanding of the experimental design.

L140-144: Authors should explain what egg rejection frequency means as it is not clear. Do Authors mean egg rejection probability (0/1). If all nests were treated with A or B treatments and all treatments were performed only once per every nest, then this variable should be named something other than “frequency”. Also, I think that the Authors should include the sex of the ejector as a fixed effect, should treat laying date and clutch size also as fixed effect (as it might alter the response) and keep nest ID as a random effect in the MCMCglmm models.

L154 and throughout the manuscript: The word “frequency” should be used only for repeated events (i.e., how frequently did one swallow evict multiple peanut shells). I think that response probability or rejection probability (not egg rejection probability, as the birds had to eject the egg and something else too) would be more suitable here, as they either rejected or accepted the mock egg and peanut shell.

L156: To make this sentence clearer, I would recommend writing plainly: “Birds treated with mock egg + peanut shell removed the peanut shell in 100% of the cases”.

L177: There was no mention of interaction terms in the description of the statistical analysis.

L184: I would advise Authors to change the colours of all figures, as there is no colour difference between the two treatment’s boxplots, neither between sexes in Figure 2. The former is not that important, but in the latter case a change of colours or an increase in contrast will help a lot.

Minor issues:

L11: In the simple summary the Authors state that nest cleaning is to avoid detection by nest predators. I agree that this could also be of an adaptive value for the evolution of this particular behavioural trait, but for the general public for which the lay summary is intended, I would recommend deleting the first part of the sentence.

L14: insert “by” before brood parasites

L15: Yes, nest sanitation and egg rejection are related behaviours, but not only because both involve cognition and occur during breeding season. There are many behaviours occurring during the breading season which involve cognition, therefore Authors should complete this sentence with some additional information. E.g., the two behaviours are connected because they impact brood survival and involve an accept/reject decision regarding the content of the nest.

L18: Not sure what the Authors mean by egg rejection occurring more “quickly”. Do they mean that it evolves faster or the action itself is performed faster than the egg rejection? The former fits better here, but then Authors should clarify the wording or rephrase this sentence. This also applies to L29 where “quickly” is used. Maybe reformulate it more explicitly as “foreign objects were removed sooner upon discovery”?

L24-26: I would argue that removal of faecal sacs and eggshells upon hatching is indeed a behaviour evolved to lower predation rates, but twigs, stones, leaves etc. have no odour to attract predators or brood parasites. I would stress out here that foreign objects might harm the eggs or nestlings, as this is a more plausible danger, and mention predation and brood parasitism only afterwards.

L148-152: Authors should include the proper citation for R and for the used R packages.

L202: I think MCMCglmm models have critical intervals instead of confidence intervals, and they are abbreviated as Cr. I. not CI.

Author Response

Reviewer1

In this study Yang et al. perform a clever experiment on barn swallow (Hirundo rustica) nests, which is an occasional host of the brood parasitic common cuckoo (Cuculus canorus). The experiment consists of placing dummy eggs with or without a peanut shell in 133 swallow nests, to test nest sanitation and egg rejection rates in the studied species. The manuscript is well written, clear, and easy to read, and it contributes to the body of knowledge regarding egg rejection decisions. I have only some easily addressable issues regarding the manuscript, which I listed bellow.

 Reply: Thank you very much for reviewing our manuscript. We have improved the manuscript according to your suggestion. Please see the manuscript revision (in yellow highlight) and the reply below.

Major issues:

L100-133: There is no information in the Data collection section about the sample size of the experiment, the sample sizes of the experimental groups, how they were distributed to one of the two groups. I found the sample sizes in Table 1, but that is only on page 4, so it should be mentioned in the main text also.

 Reply: We have added such information. Please see lines 105-107.

L100-133: Egg mass might alter rejection decisions. Authors should state the mass of the mock eggs and if it differed significantly from the mass of barn swallow eggs. If there is a significant difference in mass, authors should address this in the Discussion.

 Reply: We have clarified this. Please see lines 109-111.

L100-133: How the sex of the birds was identified and how consistent is the sex identification is missing from the text. In a species with a high degree of similarity between sexes, the repeatability of sex identification should be tested and mentioned.

 Reply: We have clarified this. Please see lines 129-133.

L100-133: I would be happy to see a picture of the eggs and peanut shell, either in the nest or outside of it. I think adding this would aid the understanding of the experimental design.

 Reply: We have added a picture for this. Please see the new Figure 1.

L140-144: Authors should explain what egg rejection frequency means as it is not clear. Do Authors mean egg rejection probability (0/1). If all nests were treated with A or B treatments and all treatments were performed only once per every nest, then this variable should be named something other than “frequency”. Also, I think that the Authors should include the sex of the ejector as a fixed effect, should treat laying date and clutch size also as fixed effect (as it might alter the response) and keep nest ID as a random effect in the MCMCglmm models.

 Reply: We have changed the frequency to probability because each nest received only one treatment. Please see lines 4, 20, 32, 69-71, 150, 152, 163, 187, 212, 223. We have included nest ID as a random effect, and clutch size and the interaction of treatment and clutch size as fixed effects. Please see lines 152-155, 185-190 and the new Table 2. For the sex effect, we cannot test it on egg rejection probability in this model because (1) sex can only be confirmed in rejection cases because acceptance cases did not involve sex effect; (2) sex was not confirmed in some cases of rejection because video record covered a limited period of 24 hr. To text the sex effect by a glmm model, only the rejection cases with sex information can be included. In such case, sex can be used as a fixed effect on the treatment (the response variable) to compare the sex ratio between two treatments. However, we believed that such simple comparison of sex ratio can be performed by Fisher’s exact test.

L154 and throughout the manuscript: The word “frequency” should be used only for repeated events (i.e., how frequently did one swallow evict multiple peanut shells). I think that response probability or rejection probability (not egg rejection probability, as the birds had to eject the egg and something else too) would be more suitable here, as they either rejected or accepted the mock egg and peanut shell.

 Reply: We have changed the frequency to probability because each nest received only one treatment. Please see lines 4, 20, 32, 69-71, 150, 152, 163, 187, 212, 223.

L156: To make this sentence clearer, I would recommend writing plainly: “Birds treated with mock egg + peanut shell removed the peanut shell in 100% of the cases”.

 Reply: We have revised this sentence. Please see lines 165-166.

L177: There was no mention of interaction terms in the description of the statistical analysis.

 Reply: We have added. Please see lines 155-156.

L184: I would advise Authors to change the colours of all figures, as there is no colour difference between the two treatment’s boxplots, neither between sexes in Figure 2. The former is not that important, but in the latter case a change of colours or an increase in contrast will help a lot.

 Reply: We have changed the colours of figure 2 (formerly figure 1, the new figure 1 was a photo of treatment) and figure 3 (formerly figure 2). Please see the new figure 2 and 3.

Minor issues:

L11: In the simple summary the Authors state that nest cleaning is to avoid detection by nest predators. I agree that this could also be of an adaptive value for the evolution of this particular behavioural trait, but for the general public for which the lay summary is intended, I would recommend deleting the first part of the sentence.

Reply: We have deleted the first part of the sentence.

L14: insert “by” before brood parasites

Reply: We have added it. Please see line 13.

L15: Yes, nest sanitation and egg rejection are related behaviours, but not only because both involve cognition and occur during breeding season. There are many behaviours occurring during the breading season which involve cognition, therefore Authors should complete this sentence with some additional information. E.g., the two behaviours are connected because they impact brood survival and involve an accept/reject decision regarding the content of the nest.

Reply: We have added this sentence. Please see lines 13-15.

L18: Not sure what the Authors mean by egg rejection occurring more “quickly”. Do they mean that it evolves faster or the action itself is performed faster than the egg rejection? The former fits better here, but then Authors should clarify the wording or rephrase this sentence. This also applies to L29 where “quickly” is used. Maybe reformulate it more explicitly as “foreign objects were removed sooner upon discovery”?

Reply: We have revised this throughout the text. Please see lines 18, 30, 75, 80, 228, 267-268, 277.

L24-26: I would argue that removal of faecal sacs and eggshells upon hatching is indeed a behaviour evolved to lower predation rates, but twigs, stones, leaves etc. have no odour to attract predators or brood parasites. I would stress out here that foreign objects might harm the eggs or nestlings, as this is a more plausible danger, and mention predation and brood parasitism only afterwards.

Reply: We have added this information. Please see lines 24-27.

L148-152: Authors should include the proper citation for R and for the used R packages.

Reply: Such information was mentioned. Please see lines 157-160.

L202: I think MCMCglmm models have critical intervals instead of confidence intervals, and they are abbreviated as Cr. I. not CI.

Reply: We have revised this. Please see lines 213-216.

Reviewer 2 Report

Review of manuscript: Egg rejection and nest sanitation in an Island population of barn swallows (Hirundo rustica): frequency, response latency, and sex effects by Yang Q et al.

The authors continue the research, by the team of Canchao Yang from College of Life Sciences, Hainan Normal University, on the cognition of behavior leading to adaptation in the system: brood parasites -host in the process of evolution. The research problem has been well defined and is a continuation of a follow-up paper published in Current Zoology 2021, 67(6) and in Animal Cognition 2015, (18).

Introduction

A research problem of interest from the point of view of looking at the relationship between exaptation and adaptation, in the context of the evolutionary process and the selection leading to the behavior of rejection the eggs of brood parasites. In the present study, the authors test the ability of recognize of foreign eggs and non-eggs objects, assuming that nest cleaning will be performed faster than egg removal in the same nest. They predict this behavior on the basis of a formulated hypothesis (Nest Sanitation Hypothesis), claiming that nest sanitation is an evolutionarily older behavioral activity, responsible for one aspect of lowering predation pressure, which has been co-opted for the behavior of nest defense against nest parasites, as a response to removing foreign eggs.

Such an aspect has been studied, admittedly in a colony-nesting species in which nest parasitism is not observed (Herring gull). Stratton and Dearborn (Current Zoology 2021, 67) discussing the results of gulls' response to nest cleaning formulate key issues that need to be known, including the commonality of a connection between nest sanitation and parasitic egg rejection, the mechanism underlying such a connection (attention vs. priming), and the duration of any impact of performing nest sanitation behavior. It is unfortunate that the Authors do not refer to this work at all, either in the introduction or in the discussion of their own results. Behavioral variation is important in the context of evolutionary lag or equilibrium.

Material and methods

Data was collected correctly. The experiments are not objectionable. Defining the recording time to 24 hours seems fully reasonable and rational, and the classification of behaviors is not objectionable. Statistical analyses are well done, although:

1/ I wondered why the MCMCglmm models used clutch size as a random variable, rather than the date of the first laid egg? Did the authors compare these different models with a different fixed and random effects?

Results

1/ One point requiring clarification or correction: The authors use Fisher's exact test to compare the ratio of females and males exhibiting egg rejection or nest sanitation (lines: 146-147), but in the results they give χ2 values and the notation Fishers' exact test (lines: 168, 172, 176), - but χ2 values and Fisher's exact test are different analytical approaches. In the case of Fisher's exact test, we only calculate the probability - please check if the p-values given in the paper are correct, do they refer to the chi-square test by any chance?

2/ Line 158: why the designation of the sum of ranks is V; usually this value is referred to as W, (sometimes T), and in the case of samples larger than 25 it is approximated by the Z test (Normal approximation for larger samples) and then we give the Z value as the value of the statistic. Normal approximation for larger samples - the same principle is used for the Wilcoxon rank sum test.

3/ Figure 2 – Is there validity to compare the proportion (%) when the samples are 19-28 elements? - isn't it better to show the number of events and possibly test the problem in a 3x3 arrangement.

In the title Island - small letter

Sincerely 

Author Response

Reviewer2

The authors continue the research, by the team of Canchao Yang from College of Life Sciences, Hainan Normal University, on the cognition of behavior leading to adaptation in the system: brood parasites -host in the process of evolution. The research problem has been well defined and is a continuation of a follow-up paper published in Current Zoology 2021, 67(6) and in Animal Cognition 2015, (18).

 Reply: Thank you very much for reviewing our manuscript. We have improved the manuscript according to your suggestion. Please see the manuscript revision (in yellow highlight) and the reply below.

Introduction

A research problem of interest from the point of view of looking at the relationship between exaptation and adaptation, in the context of the evolutionary process and the selection leading to the behavior of rejection the eggs of brood parasites. In the present study, the authors test the ability of recognize of foreign eggs and non-eggs objects, assuming that nest cleaning will be performed faster than egg removal in the same nest. They predict this behavior on the basis of a formulated hypothesis (Nest Sanitation Hypothesis), claiming that nest sanitation is an evolutionarily older behavioral activity, responsible for one aspect of lowering predation pressure, which has been co-opted for the behavior of nest defense against nest parasites, as a response to removing foreign eggs.

Such an aspect has been studied, admittedly in a colony-nesting species in which nest parasitism is not observed (Herring gull). Stratton and Dearborn (Current Zoology 2021, 67) discussing the results of gulls' response to nest cleaning formulate key issues that need to be known, including the commonality of a connection between nest sanitation and parasitic egg rejection, the mechanism underlying such a connection (attention vs. priming), and the duration of any impact of performing nest sanitation behavior. It is unfortunate that the Authors do not refer to this work at all, either in the introduction or in the discussion of their own results. Behavioral variation is important in the context of evolutionary lag or equilibrium.

  Reply: We have added this reference to the manuscript. Please see lines 224-225, 359-360.

Material and methods

Data was collected correctly. The experiments are not objectionable. Defining the recording time to 24 hours seems fully reasonable and rational, and the classification of behaviors is not objectionable. Statistical analyses are well done, although:

1/ I wondered why the MCMCglmm models used clutch size as a random variable, rather than the date of the first laid egg? Did the authors compare these different models with a different fixed and random effects?

 Reply: We have improved the model according to your and another reviewer’s suggestion. Please see lines 152-156.

Results

1/ One point requiring clarification or correction: The authors use Fisher's exact test to compare the ratio of females and males exhibiting egg rejection or nest sanitation (lines: 146-147), but in the results they give χ2 values and the notation Fishers' exact test (lines: 168, 172, 176), - but χ2 values and Fisher's exact test are different analytical approaches. In the case of Fisher's exact test, we only calculate the probability - please check if the p-values given in the paper are correct, do they refer to the chi-square test by any chance?

 Reply: The values we presented were the odd ratios (OR) rather than χ2; we have deleted them.

2/ Line 158: why the designation of the sum of ranks is V; usually this value is referred to as W, (sometimes T), and in the case of samples larger than 25 it is approximated by the Z test (Normal approximation for larger samples) and then we give the Z value as the value of the statistic. Normal approximation for larger samples - the same principle is used for the Wilcoxon rank sum test.

  Reply: Both the Wilcoxon rank sum test and Wilcoxon signed-rank test were used in the analyses. According to the output form R software, the designation W and V were presented for these two tests, respectively. I have changed V to Z as Z was more common. Please see line 167.

3/ Figure 2 – Is there validity to compare the proportion (%) when the samples are 19-28 elements? - isn't it better to show the number of events and possibly test the problem in a 3x3 arrangement.

 Reply: It was incapable to use 3x3 because the data belong to different groups that were not comparable. (a): for the nests both egg rejection and nest sanitation occurred in mock + peanut trial; (b): for the nests only egg rejection occurred in mock trial; (c): for the nests only nest sanitation occurred in mock + peanut trial. Therefore, sex ratio was compared within groups rather than between groups.

In the title Island - small letter

 Reply: We have revised it. Please see line 3.